# Design Optimization of a Phototherapy Extracorporeal Membrane Oxygenator for Treating Carbon Monoxide Poisoning

**DOI:** 10.3390/bioengineering10080969

**Published:** 2023-08-16

**Authors:** Edidiong Etim, Anastasia Goulopoulos, Anna Fischbach, Walfre Franco

**Affiliations:** 1Department of Biomedical Engineering, University of Massachusetts Lowell, Lowell, MA 01854, USA; 2Department of Anesthesiology, University Hospital, 52074 Aachen, Germany; 3Wellman Center for Photomedicine, Massachusetts General Hospital, Boston, MA 02114, USA; 4Department of Dermatology, University of Massachusetts Chan Medical School, Worcester, MA 01655, USA

**Keywords:** blood phototherapy, carbon monoxide poisoning, computational modeling, computational fluid dynamics (CFD), extracorporeal membrane oxygenation, hyperbaric oxygen, photodissociation

## Abstract

We designed a photo-ECMO device to speed up the rate of carbon monoxide (CO) removal by using visible light to dissociate CO from hemoglobin (Hb). Using computational fluid dynamics, fillets of different radii (5 cm and 10 cm) were applied to the square shape of a photo-ECMO device to reduce stagnant blood flow regions and increase the treated blood volume while being constrained by full light penetration. The blood flow at different flow rates and the thermal load imposed by forty external light sources at 623 nm were modeled using the Navier-Stokes and convection–diffusion equations. The particle residence times were also analyzed to determine the time the blood remained in the device. There was a reduction in the blood flow stagnation as the fillet radii increased. The maximum temperature change for all the geometries was below 4 °C. The optimized device with a fillet radius of 5 cm and a blood priming volume of up to 208 cm^3^ should decrease the time needed to treat CO poisoning without exceeding the critical threshold for protein denaturation. This technology has the potential to decrease the time for CO removal when treating patients with CO poisoning and pulmonary gas exchange inhibition.

## 1. Introduction

Carbon monoxide (CO) is a colorless, odorless, and tasteless gas [1]. CO poisoning is responsible for up to 50,000 emergency department visits each year as a result of intentional and unintentional CO exposure [2]. Acute medical expenses and lost earnings associated with accidental CO poisoning have led to an annual economic burden of over $1.3 billion in the United States [3]. Formed by the incomplete combustion of organic compounds, sources of CO include fires, the incomplete combustion of fuels, using a burner, heating or cooking with insufficient ventilation, exhaust gas, smoke from cigarettes, and industrial accidents [1,4].

Hemoglobin (Hb) has a 200 to 250-fold higher affinity to CO than to oxygen. Therefore, CO exposure reduces the oxygen-carrying capacity of Hb. CO also binds to other heme-containing proteins, such as cytochromes C oxidase, which leads to an inhibition of mitochondrial respiration [4]. CO-mediated reductions in oxygen delivery and the inhibition of cytochrome C oxidase cause various symptoms, including headache, nausea, vomiting, blurred vision, dizziness, chest pain, shortness of breath, altered mental status, cardiac dysrhythmia, cerebral infarction, pulmonary edema, and coma [4].

The current therapy for CO poisoning is breathing 100% oxygen at normobaric or hyperbaric pressure [5]. The CO elimination half-life (HbCO-t_1/2_) is approx. 4 to 5 h when breathing air [6], 1.5 to 2.5 h when breathing 100% oxygen with a mask at a normal atmospheric pressure (NBO), and less than 30 min using hyperbaric oxygen (HBO) at 2.5 atm [7]. The treatment using normobaric oxygen is commonly employed in mild cases of CO poisoning, while treatment using hyperbaric oxygen is used in more severe cases [7]. There are only a limited number of hyperbaric oxygen chambers in the United States [5]. Since patients need to be transported to a hyperbaric therapy center, there is often a delay in the delivery of HBO. CO poisoning can be associated with acute lung injury caused by smoke inhalation [8]. In this case, HBO therapy is less effective since the gas exchange through the lungs is impaired.

Extracorporeal membrane oxygenators (ECMOs) are used to mechanically ventilate the blood when heart or lung functions are impaired. Oxygenators have evolved over the years. The first oxygenators, a bubble oxygenator and a rotating disc, were built in 1950 [9,10]. However, trauma to the blood was observed due to the direct exposure of the blood to the gas. In 1944, Kolff and Berk observed that blood was arterialized as it passed through the cellophane chambers of the artificial kidney, which led to the development of plate-type membrane oxygenators with sheet membranes by G.H.A Clowes, Jr. and the coil-type oxygenator with polyethylene membranes by W.J. Kolff in 1956 [11]. The design of oxygenators has mainly focused on the surface interactions and gas transfer capabilities [9]. Modifications to the membranes range from the polymer type (including polypropylene, aromatic polyimide, poly-4-methylpentene-1, polydimethylsiloxane, and fluorinated polyamide), method of fabrication, and fiber winding, spacing, and coating to increase the hemocompatibility, mechanical strength, and mass transfer performance [12].

The shape of oxygenators has predominately been influenced by the arrangement of the membranes and the passage of the blood through the device. In order to promote the mixing of blood, Kolobow created the spiral membrane lung, which was a long flat membrane wrapped around a central cylinder [9]. Current oxygenators with slight modifications have been built using this spiral-wound-type cylindrical form [13], such as the Medos Hilite 2400 LT, Eurosets ECMO Pediatric and New Born, Sorin Lilliput 2, Sorin EOS ECMO, Chalice Paragon^PMP^ Pediatric, Infant and Neonatal devices [12]. The plate-type membrane oxygenator is square-shaped and was adopted by the commercially available Maquet Quadrox-iD oxygenator.

The ECMO design is still progressing to improve patient care. However, the traditional parameter and shape optimization of an ECMO device can be costly and time consuming since numerous physical experiments must be performed. Alternatively, computational fluid dynamics (CFD) can provide a fast, flexible, and accurate model [14]. CFD has been used to model blood flow in pumps, gas flow in gas exchange hollow fibers, and hemodynamic performance with good agreement with ex vivo and in vitro experiments [14]. Han et al. analyzed and compared three leading centrifugal blood pumps to identify the risk of hemolysis when using CFD and experiments [15]. Tang et al. reported various 2D and 3D CFD models to evaluate the gas exchange with varied blood flow conditions (steady and pulsating flow) and membrane representations (porous media and fiber membranes) [16]. Turri et al. developed a 2D numerical simulator to model the mass transfer of oxygen in hollow fiber membranes. They evaluated the local pH value and conditions in which a mass transfer occurred across the full length of the membrane [17]. Messai et al. developed a numerical model to understand the role of various determinants on oxygenation during veno-venous ECMO therapy [18].

Similarly, Conrad et al. used CFD to evaluate the factors that could influence recirculation during two-site veno-venous ECMO therapy. They found that the minimal influence factors of recirculation were the atrial blood volume, the position of the cannula in the inferior vena cava relative to the cavo-atrial junction, the number of side holes in the return cannula, and the blood viscosity. The moderate influence factors of recirculation were the heart rate, the diameter of the return cannula, and the extracorporeal blood flow. The major influence factors of recirculation were the superior vena cava to inferior vena cava blood flow ratio, the position of the superior vena cava relative to the cavo-atrial junction, and the orientation of the return cannula relative to the cavo-atrial junction [19]. Stevens et al. performed a steady-state and transient CFD simulation using a patient-specific geometrical model of the aorta to quantify the relationship of the ECMO support level with the mixing zone and left ventricle flows [20]. Nezami et al. conducted a CFD analysis to quantify hemodynamics and perfusion at varying levels of ECMO blood flow to simulate the different degrees of heart failure [21]. Fragomeni et al. compared the central and peripheral cannulation with CFD and showed that central cannulation was less thrombogenic [22]. Despite certain limitations, CFD allowed for dynamic and progressive design modeling using observations that could have been difficult to conduct experimentally.

In 1986, Haldane and Smith reported the dissociation of CO, not oxygen but from Hb using visible light [23]. In a previous study, we developed an extracorporeal veno-venous membrane oxygenator that facilitated the exposure of visible light to blood (photo-ECMO device) [24]. Table 1 shows the published development of the photo-ECMO device over the last four years.

Recently, we conducted a numerical study to analyze the light penetration, blood flow distribution, and heat generation of the blood within a photo-ECMO device prototype that was successfully used in experiments to remove CO from blood using light at three different wavelengths [25,27]. The numerical study showed that light at 620 nm (red light) relative to 460 and 523 nm was the optimal wavelength for CO removal in the device as it maintained a blood temperature below thermal damage and penetrated deeper compared to the other wavelengths. However, stagnation regions within the device resulted in hot spots, which were more evident for shorter light wavelengths. Additionally, the photo-ECMO device prototype in the larger animal study, the “maxi-ECMO”, exhibited a blood volume of 60 ${\mathrm{cm}}^{3}$. Experimentally, six devices were assembled in parallel to increase the treated blood volume to 360 ${\mathrm{cm}}^{3}$ [24]. This number of devices was bulky, inefficient, and clinically not feasible.

The objective of this study was to optimize the shape of the photo-ECMO device for clinical use with blood volumes of 150 to 320 ${\mathrm{cm}}^{3}$ that are commonly used in ECMO devices, while reducing the blood stagnation areas using CFD. The scaled blood treatment volume of the photo-ECMO device, presented here, is meant to effectively treat human adults with CO poisoning using one device.

## 2. Study Design and Methods

The light propagation, blood flow, and heat generation within the photo-ECMO device were modeled using the Monte Carlo method, laminar Navier-Stokes, and the convection–diffusion equations, respectively.

### 2.1. The Photo-ECMO Device

The photo-ECMO device (Figure 1) comprised microporous polypropylene membranes for the gas exchange and a clear plexiglass encapsulation for the light penetration. The polypropylene hollow fiber mats were stacked 90° relative to each layer and enclosed in a plexiglass casing with an 11 mm thickness. This thickness was chosen during the preliminary in vitro experiments comparing the irradiance with devices of varying thickness (11 mm, 9 mm, and 5 mm). The blood flowed within the chamber around the hollow fibers. However, the fibers were not modeled in this study for the sake of computational simplicity.

### 2.2. Light Propagation

The Monte Carlo simulations were used to model light transport in the tissues. The Monte Carlo packages, MCML and CONV, used probability distributions to describe the step size of the photons as they interacted with the tissue and the angle of deflection in the scattering events [29]. The light propagation was previously modeled using the Monte Carlo method [27]. Briefly, 10 million photons were launched into a 1.0 cm × 1.5 cm grid with a resolution of 0.001 cm. This domain was used to prevent boundary effects. The beam radius was 0.33 cm with an input energy of 0.036 J at a 620 nm wavelength. These parameters were calculated using the LED specifications from the LEDs that were used in the in vitro and in vivo studies [24,25]. The light propagation in the plexiglass was not modeled. However, the plexiglass had a minimum light transmission of 92%, so we measured the transmitted light through an 11 mm thick plexiglass sheet using a power meter. The measured value was used as the surface irradiation on the blood pool.

The LEDs were modeled as low-intensity lasers, despite emitting light with a viewing angle that was uncharacteristic of lasers. A gaussian beam was used instead of a circularly flat beam to better mimic the spatial variation characteristic of LED light deposition. Although it did not account for the photon incident at an angle from the surface, the radius of the beam was calculated using the LED’s path length and viewing angle to improve the approximation. An additional limiting factor of the current model was the omission of the gas exchange membranes, although we showed in a previous study that it only contributed to scattering, not to the absorption of the photons [27].

The heat generation within the blood was due to the absorption of the photons by chromophores. Herein, we only modeled HbCO, assuming the most extreme case for treatment when the blood was fully poisoned. During photodissociation, other chromophores (HbO and Hb) were present. However the absorption properties of HbO and Hb (1.12 cm^−1^ and 6.12 cm^−1^, respectively) [27] at 620 nm were on the same order of magnitude as the absorption properties of HbCO (1.56 cm^−1^). Given they had the same scattering coefficient, we were able to simplify this model to a pool comprising only HbCO, expecting slight variations in energy deposition.

### 2.3. Modeling Geometries

Three designs were analyzed in this study. The first simulated geometry had a blood chamber of 20 cm × 20 cm × 1 cm, as shown in Figure 2A. The device had inlet and outlet tubes with inner and outer diameters of 0.635 cm and 0.655 cm, respectively, and a height of 1.4 cm from the outer plexiglass surface. The inlet and outlet were positioned 12 mm from the edge. The device had 20 LEDs at 620 nm at the top and at the bottom.

To modify the square geometry, 5 and 10 cm radii fillets were applied to all the corners. MATLAB was used to calculate the inlet and outlet position after the fillets were added. The new position was calculated by adding the fillet radius multiplied by a scaling factor to the initial distance of 12 mm from the corners. The resulting geometries are shown in Figure 2B and Figure 2C, respectively.

COMSOL Multiphysics (Burlington, MA, USA) was used to generate the computational mesh, as shown in Figure 2. The elements were created using a free tetrahedral mesh. The meshing of the blood compartment domain (including the inlet tubes) was adjusted for the modeling fluid dynamics, and the plexiglass domain was meshed for general physics. Eight boundary layers with a stretch factor of 1.2 were generated on non-slip surfaces (inner boundary of the plexiglass casing). Table 2 shows the type of mesh used for each geometry. The total number of elements for Geometries 1, 2, and 3 were 4,640,396 elements, 3,153,170 elements, and 3,270,021 elements, respectively.

### 2.4. Numerical Analysis

The governing equations for flow and heat were solved numerically using the commercial finite element analysis software COMSOL Multiphysics 6.0 with MATLAB in an Intel(R) Xeon(R) W-2133 with 64.0 GB RAM and 3.60 GHz. The iterative solver, GMRES, was used with a relative tolerance of 1E-3, 1E-5, and 1E-2 for flow, particle tracing, and heat, respectively. An initial time step of 0.06 s (0.001 min) was chosen by default in the solver based on the physics combination. An adaptive time stepping scheme was also used to maintain the desired relative tolerance.

Prior to this study, grid sensitivity analyses were performed to evaluate the effect of the mesh refinement on the solution. The meshes were created from coarse to extremely fine. Table 3 shows the change in the solution with a mesh refinement for Geometry 1 at 0.25 L/min. There was no significant change in the results with the mesh refinement. However, in order prevent the influence of the mesh refinement on the particles lingering within the device, the finest mesh that could be converged by the system was used.

### 2.5. Blood Flow

The blood flow in the photo-ECMO device was modeled in 3D. The blood was modeled as a non-Newtonian fluid using the steady-state Navier-Stokes Equation (1) and continuity Equation (3).
(1)$$\rho \left(u\;\cdot \nabla \right)u=\nabla \cdot \left[-pI+\kappa \right]+F,$$
(2)$$\kappa =\mu \left(\nabla u+{\left(\nabla u\right)}^{T}\right),$$
(3)ρ∇⋅u=0,
where ρ is the density of the blood (1057 ${\mathrm{Kg}/m}^{3})$, *u* is the velocity, μ is viscosity, and *p* is the pressure. The first term in the right operand of Equation (1) accounts for the viscous forces and is further defined in Equation (2). *F* is any external force applied to the fluid, which for this study was zero.

The Carreau inelastic model was used for modeling the blood as a non-Newtonian fluid [19].
(4)$$\mu =\mu_{\infty }+\left(\mu_{0}-\mu_{\infty }\right){\left[1+{\left(\lambda \dot{\gamma }\right)}^{2}\;\right]}^{\frac{n-1}{2}},$$
where μ is the apparent shear rate, $\mu_{\infty }$ is the infinite shear rate (0.0035 Pa⋅s), $\mu_{0}$ is the zero shear rate (0.056 Pa⋅s), λ is the relaxation time (3.313 s), *n* is the power index (0.3568), and $\dot{\gamma }$ is the shear rate calculated by $\sqrt{2S\;:S}$, $S=\frac{1}{2}\;\left[\nabla u+{\left(\nabla u\right)}^{T}\right]$.

The blood flowed in at three different flow rates: 0.25, 1, and 3 L/min, respectively, which were within the range of flow rates used in commercial ECMO devices (up to 7 L/min) [12]. The outlet boundary condition was set to the atmospheric pressure.

### 2.6. Particle Tracing

The particle position was computed using Newton’s second law (Equation (5)).
(5)$$\frac{d\left(m_{p}v\right)}{dt}=F_{t},$$
where *m_p_* is the particle mass, *v* is the particle velocity, *t* is the time, and *F_t_* is any external force acting on the particle, in this case the drag force.

One hundred particles were launched into the photo-ECMO device to estimate the time it would take for the blood particles to pass through the different geometries at blood flow rates of 0.25, 1, and 3 L/min. A drag force node was included since the Newtonian formulation solved for the particle velocity with the assumption that all the other forces were perfectly counterbalanced by the drag force at any point in time. All the particles were simultaneously released at the start of the simulation with their velocities dependent on the velocity field solved at the corresponding blood flow rate. The simulation was run for 15 min and the values were stored every minute.

### 2.7. Blood Heating

Blood heating occurs due to the deposition of photons during irradiation. The effect of this was modeled using the transient convection–diffusion equation.
(6)$$\rho c_{p}\frac{\partial T}{\partial t}+\rho c_{p}u\cdot \nabla T+\nabla \cdot \left(-\kappa \nabla T\right)=\mu_{a}\phi ,$$
where $c_{p}$ is the specific heat (3600 J/(Kg K)), *T* is the blood temperature, *t* is the time, *u* is the velocity, κ is the thermal conductivity (0.55 W/(m K)) [30], and ϕ is the fluence. The fluence for each LED was solved using the Monte Carlo simulation, described above. Considering that the maximum temperature did not exceed 10 °C [3], the temperature dependent thermal properties were not considered.

The thermal boundary condition was the natural convection between the external boundary of the plexiglass and the air.
(7)$$-\kappa \nabla T\cdot n=h\left(T_{\infty }-T\right),$$
where *h* is the heat transfer coefficient for a vertical/horizontal hot plate evaluated in COMSOL. The specific correlation can be found in [31,32]. The thermal initial condition used was room temperature at 20℃. The heat transfer in the plexiglass was solved using Equations (6) and (7). However, there was neither convection (*u* = 0) nor light absorption in the transparent solid ($\mu_{a}\phi$ = 0) and the equation was simplified to $\rho c_{p}\frac{\partial T}{\partial t}+\nabla \cdot \left(-\kappa \nabla T\right)=0$. The density, specific heat, and thermal conductivity used for the plexiglass were 1190 Kg/m^3^, 1470 J/(Kg K), and 0.18 W/(m K), respectively.

The material property was available in the COMSOL material library. A prior computational analysis showed that as the blood flowed from the patient through a 3 ft blood inlet tube, it cooled from body temperature to room temperature [27].

## 3. Results

### 3.1. Light Propagation

The distribution and maximum intensity of the deposited energy in the blood is shown Figure 3. The deposition of the photons resulted in heat generation. Figure 3A shows the 2D light deposition in a pool of blood. The 3D fluence of one LED, as shown in Figure 3B, was derived by revolving the photon deposition 360° in MATLAB. A total of 20 LEDs on the top and bottom of the device were spaced 5 cm apart. Figure 3C,D show the distribution of the top and bottom light sources at a depth of 0.4905 cm and 0 cm, respectively. The intensity was at a maximum at y = 0.4905 cm for the top LEDs because this was where the deposition began. It was the least at y = 0 cm for the bottom LEDs because the photon deposition was reduced.

### 3.2. Blood Flow

Figure 4 shows the velocity field as the blood flowed through the three membrane oxygenators at a constant flow rate of 0.25 L/min. The arrows denoted the direction of blood flow as well as the magnitude at the location. The arrow vectors were scaled by 1000 to improve their visibility. The blood entered at a higher velocity relative to the rest of the oxygenator due to the cross-sectional change. To properly observe the velocity distribution in the photo-ECMO device, the velocity scale was reduced to a 10 mm/s as the average velocity was ~3 mm/s for all the geometries at that flow rate. A minimum of 1 mm/s was chosen to observe the variation in the stagnation regions for all the geometries.

The arrows shown at the edge of Figure 4A opposite to the inlet and outlet tubes were very small and difficult to see due to the low velocity. Additionally, some areas did not exhibit any arrows for showing the presence of stagnant regions. However, these areas were reduced in Figure 4B and were nonexistent in Figure 4C. A similar distribution was observed for the two other flow rates (not shown). Table 4 summarizes the average and maximum velocity in the inlet and outlet tubes as well as the oxygenator for the three different blood flow rates. The pressure drop was similar at the different flow rates and geometries, although Geometry 1 showed a higher pressure drop (30 Pa) at 3 L/min than Geometries 2 and 3.

### 3.3. Particle Tracing

One hundred blood particles were launched from the inlet of each geometry. The travel time to reach the outlet was recorded for all the geometries in a 10-min time frame (Figure 5). By 10 min, over 90% of the particles reached the outlet at 0.25 L/min for all the geometries. However, as the flow rate increased, the number of particles reaching the outlet within 10 min decreased. For Geometry 1, 95 particles, 84 particles, and 69 particles reached the outlet after 10 min at a blood flow rate of 0.25 L/min, 1 L/min, and 3 L/min, respectively. A similar trend was observed for Geometries 2 and 3. However in Geometry 3, the particles reached the outlet faster compared to the other geometries. After 1 min, 36 particles, 45 particles, and 64 particles reached the outlet for Geometries 1, 2, and 3, respectively, at a blood flow rate of 0.25 L/min. By the second minute, the number of particles at the outlet were similar regardless of the geometry (81, 84, and 85 for Geometries 1, 2, and 3, respectively). The number of particles at the outlet did not change for all the geometries and flow rates after 5 min. Nevertheless, the number of particles at the outlet for Geometries 1 and 2 increased by one at 0.25 L/min after 6 min, after which it remained the same.

At 0.25 L/min, Figure 5A shows there was not much of a difference when comparing the three geometries and the number of particles at the outlet. However, as shown in Figure 5B, Geometries 2 and 3 resulted in a higher number of particles at the outlet with a blood flow rate increased to 1 L/min. As shown in Figure 5C, more particles reached the outlet for Geometry 2 than Geometries 1 and 3. However, there was a considerable decrease in the number of particles reaching the outlet at 3 L/min.

### 3.4. Heat Generation

Figure 6A shows the velocity streamlines in Geometry 2 as well as the 20 top red light sources. The streamlines represent the path of the blood flow within the device. Photodissociation was instantaneous, so with the current distribution, the blood flowing through had a higher chance of being exposed to light. There was no considerable increase in the blood temperature for all the geometries. Figure 7 shows the iso-surfaces of the temperature for all the geometries.

Table 5 shows the average and maximum temperatures in the device at a steady state. The maximum temperature within the channel increased by 3.03 °C for Geometry 1 at 1 L/min, 2.19 °C for Geometry 2 at 0.25 L/min, and 1.83 °C for Geometry 3 at 1 L/min. However, the increase in the average temperature at the outlet only decreased within a degree for all the geometries and blood flow rates. The regions of high temperature (Figure 6B–D) coincided with the regions of stagnant blood flow.

## 4. Discussion

In the previous studies, we developed a “maxi” photo-ECMO device for the treatment of CO poisoning [24]. The dimensions of the blood compartment were 16 cm × 16 cm × 0.4 cm with a priming volume of 60 mL and a gas exchange area of 0.6 m^2^. The photo-ECMO system (comprising six maxi photo-ECMO devices assembled in parallel) illuminated with red light and ventilated with 100% oxygen was twice as efficient when compared to the ventilation of the photo-ECMO system with 100% oxygen alone, reducing the HbCO-t_1/2_ from 19.2 ± 4.7 min to 6.3 ± 1.2 min in vitro. In a CO-poisoned pig, the addition of phototherapy to the photo-ECMO system decreased the HbCO-t_1/2_ from 21.6 ± 2.6 min (ventilation with 100% oxygen alone) to 13.9 ± 0.4 min [24]. The study affirmed the feasibility and efficacy of the photo-ECMO device. However, although effective, the bulkiness of six devices was inefficient and clinically infeasible.

In this study, we focused on optimizing one crucial aspect of the photo-ECMO system—the geometry of the photo-ECMO device. We created a more clinically feasible device with a lower risk of complications due to the stagnation areas and bulkiness when using multiple devices. First, we increased the volume of blood in the blood compartment by modifying the surface area. However, with a constraint in the thickness of the device by the penetration of red light in the blood (~ 0.5 cm), the length, width, and height of the device was increased to 20 cm × 20 cm × 1 cm. Commercial oxygenators have a blood volume ranging from 150 ${\mathrm{cm}}^{3}$ to 320 ${\mathrm{cm}}^{3}$ for both pediatric and adult care [12]. The volume of the current model was 400 ${\mathrm{cm}}^{3}$ for Geometry 1. However, if the membranes were considered, Oxyphan 50/280 mats with a fiber density of 20 fibers per 10 mm, the estimated blood volume would be approx. 219 ${\mathrm{cm}}^{3}$, 208 ${\mathrm{cm}}^{3}$, and 171 ${\mathrm{cm}}^{3}$ for Geometries 1, 2, and 3, respectively, for the 20 membrane layers. Compared to the 360 ${\mathrm{cm}}^{3}$ priming blood volume for the six-device photo-ECMO system, one or two devices would be sufficient for either Geometries 2 or 3 to remain within the range of the required treatment volume for patients. This enables its use in ambulances at the scene of an accident. Furthermore, reducing the photo-ECMO system’s priming volume from 360 mL would lower the risk of hemodynamic changes in the patient, improving its benefits for clinical application.

The second step to create a more clinically feasible device was reducing the stagnation regions. The stagnant regions were no blood flow zones or regions where the blood flow velocity was very low. These regions were usually formed around sharp corners, as shown in Figure 4A for Geometry 1. The stagnation regions impeded the efficacy of CO removal. Thrombosis was typically associated with stagnant regions and generally led to an inefficient gas exchange due to a lack of convection [33]. Square-shaped oxygenations have been shown to result in stagnation areas with complications of thrombosis [34]. Gartner et al. showed that regions with a low blood velocity in membrane oxygenators predicted using CFD corresponded to clinical thrombotic depositions [35]. Using CFD, Conway et al. identified the stagnant flow condition in a Quadrox D (Getinge AB, Gothenburg, Sweden) oxygenator and computed tomography angiography (CTA) to identify the blood flow patterns associated with thrombus formation. Forty-one clinically used oxygenators were then divided into sections and analyzed. The analysis showed an agreement between the explanted oxygenator blood clot burden map with the predicted map. The largest blood clot burden was observed at the corners opposite the blood inlet and outlet [36]. This result was similar to the blood flow field for Geometry 1 in this study.

The effectiveness of round edges was not shown experimentally in this study. However, a similar analysis was shown by Hesselmann et al. who applied 3D potting to fill stagnant regions to allow blood to flow naturally from the inlet to the outlet of the device [33]. Instead of potting in this study, the stagnation regions were ‘cut’ out by adding fillets of different radii, leaving a shape that naturally followed the blood streamlines and improved the blood flow. The stagnant regions were less convective and resulted in an increased thermal load. The particle tracing data showed that after 5 min, the number of particles reaching the outlet did not change. Hence, we evaluated the maximum temperature of the blood in the device after 5 min. The computed values were 2.30 °C, 1.94 °C, and 1.57 °C for Geometries 1, 2, and 3 at flow rates of 3 L/min, 1 L/min, and 1 L/min, respectively. The maximum temperature decreased with an improvement in the blood flow in the device by the reduction in the stagnant regions. In the case where the blood remained in the device for longer, we computed what the maximum temperature would be at a steady state. For Geometries 1, 2, and 3, the maximum temperature increase within the oxygenator was 3.99 °C, 2.19 °C, and 2.53 °C at a blood flow rate of 1 L/min, 0.25 L/min, and 1 L/min, respectively. In the pig study, we observed a body temperature increase of 1.2 °C ± 0.2 °C when the photo-ECMO system was exposed to light [24]. With a maximum increase of approx. 4 °C, the protein denaturation did not occur [37] due to the forty external light sources. This further added to the feasibility of the photo-ECMO device for treating CO-poisoned patients.

Although there was a difference in the maximum temperature in the different geometries, the particle tracing data showed that modifying the shape did not significantly change the particle residence time in the device as expected. While more particles reached the outlet faster in Geometry 3, the total number of particles at the end of the simulation remained similar for the three geometries. Moreover, as the flow rate increased, more particles reached the outlet in Geometry 2 compared to the other geometries. Hence, Geometry 2, with the 5 cm fillet radius, was more optimal compared to Geometry 3 since there was a reduction in the stagnation areas (per the particle tracing results), and its priming volume of 208 ${\mathrm{cm}}^{3}$ was higher than Geometry 3, 171 ${\mathrm{cm}}^{3}$.

Our previous experimental studies outlined in Table 1 demonstrated the increase in CO elimination using phototherapy as well as hyperoxygenation. A photo-ECMO system that consisted of four hyperbaric photo-ECMO devices (ventilated with 100% oxygen at 1.33 atm) further decreased HbCO-t_1/2_ compared to the photo-ECMO system composed of six devices that were ventilated with 100% oxygen at an ambient pressure (5.2 ± 0.4 min vs. 6.3 ± 1.2 min) [26]. These results showed that an increase in the oxygen pressure reduced the number of photo-ECMO devices needed [26]. Additionally, we integrated light-diffusing optical fibers for CO elimination in a small unit-cell ECMO device and established a non-parametric mathematical model for CO elimination [28]. Red and blue lasers were used at multiple power ranges. For all the designs, the half-life of HbCO was less than 10 min.

A limitation to this study was not modeling CO removal. However, the aim of this work was to guide the building of a photo-ECMO device for clinical use. In the next step, we aim to build Geometries 2 and 3 and test them using CO-poisoned blood in vitro. Based on the results of these experiments, we aim to develop a mathematical model similar to the one we established in a previous study to predict CO removal [28]. By establishing a mathematical model, we can calculate CO removal under various scenarios without the need for a high number of in vitro experiments. Some of the parameters we aim to explore include the number of gas exchange layers, the gas inlets and outlets, the gas flow direction, the number of light sources, and the different blood flow rates.

Further geometry modifications can be explored to improve the flow and gas transfer in the photo-ECMO device. Current oxygenators have a cylindrical shape. The photo-ECMO’s oxygenator can be designed cylindrically with light sources arranged around it. A limitation to the diameter of this cylinder would be the penetration of light through the blood compartment of the device. Alternatively, the cylindrical shape could be modified by illuminating it from the core of the cylinder, e.g., by using a cross-sectional ring or donut geometry with LEDs to irradiate the inside and outside diameters. In the future, we can also use devices that are already optimized for blood flow and incorporate light, either by LEDs or optical fibers, with an increased oxygen pressure for improved results.

## 5. Conclusions

Modifications to the square shape of the photo-ECMO device, by adding a fillet radius of 5 cm, improved the blood flow and decreased the stagnation regions that could result in blood clotting. The optimized device was more compact and efficient with a blood volume up to 208 ${\mathrm{cm}}^{3}$ and without the risk of thermal blood damage due to forty external light sources with a wavelength of 623 nm. This technology has the potential to decrease the time for CO removal using one or two devices, when treating patients with CO poisoning and pulmonary gas exchange inhibition.

## Figures and Tables

**Figure 1 bioengineering-10-00969-f001:**
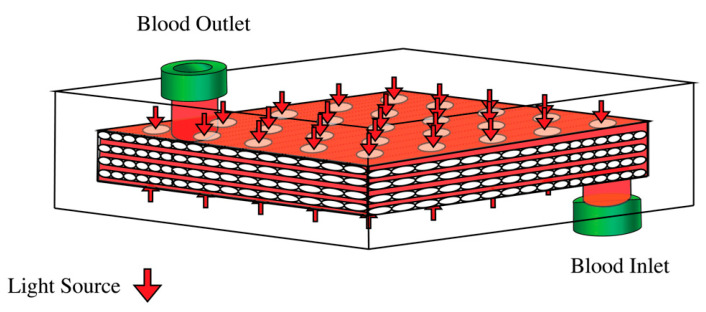
The photo-ECMO device. Polypropylene hollow fiber mats, stacked 90° relative to each layer, were enclosed in clear plexiglass with light sources above and beneath. The arrows denote the location of the light sources.

**Figure 2 bioengineering-10-00969-f002:**
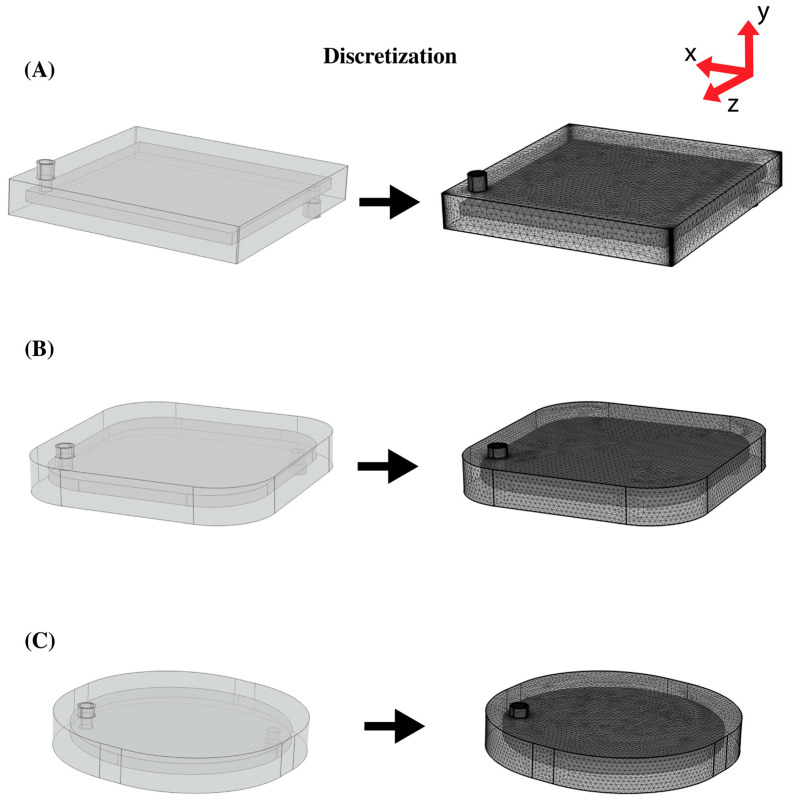
Modeling geometries and corresponding discretization. (**A**) Geometry 1, (**B**) Geometry 2, and (**C**) Geometry 3. The inner domain is the blood compartment filled with blood, and the outer domain is the plexiglass case. The blood entered the photo-ECMO device through a blood inlet on the bottom of the device and exited the photo-ECMO device through a blood outlet on the top of the device.

**Figure 3 bioengineering-10-00969-f003:**
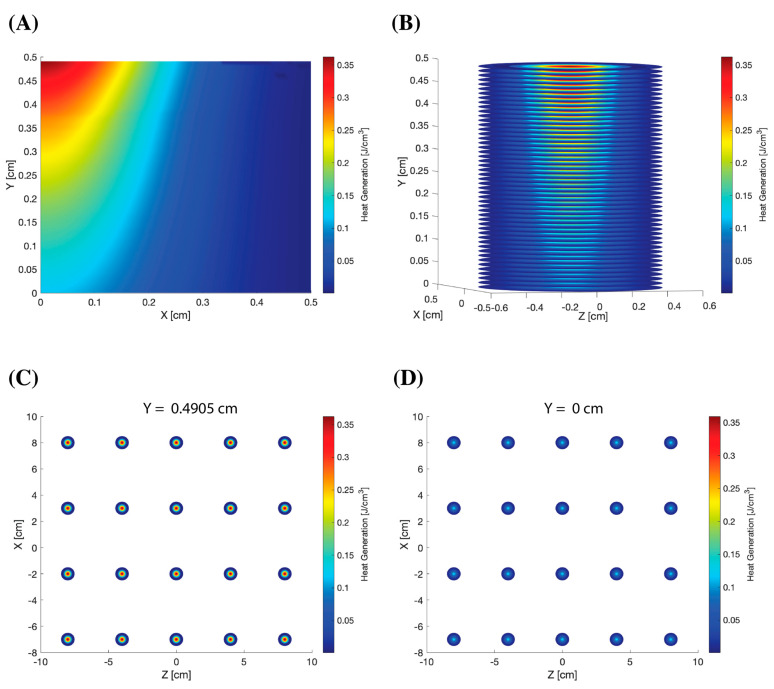
(**A**) 2D light deposition in a pool of blood, 1 cm × 1.5 cm. (**B**) 3D light deposition derived by revolving the 2D light deposition 360° in MATLAB. (**C**) Distribution of 20 red light sources at the top of the photo-ECMO device (Y = 0.4905 cm). (**D**) Distribution of 20 red light sources at the bottom of the photo-ECMO device (Y = 0 cm). A Gaussian beam with a radius of 0.33 cm and input energy of 0.035 J at a penetration depth of 0.5 cm were used.

**Figure 4 bioengineering-10-00969-f004:**
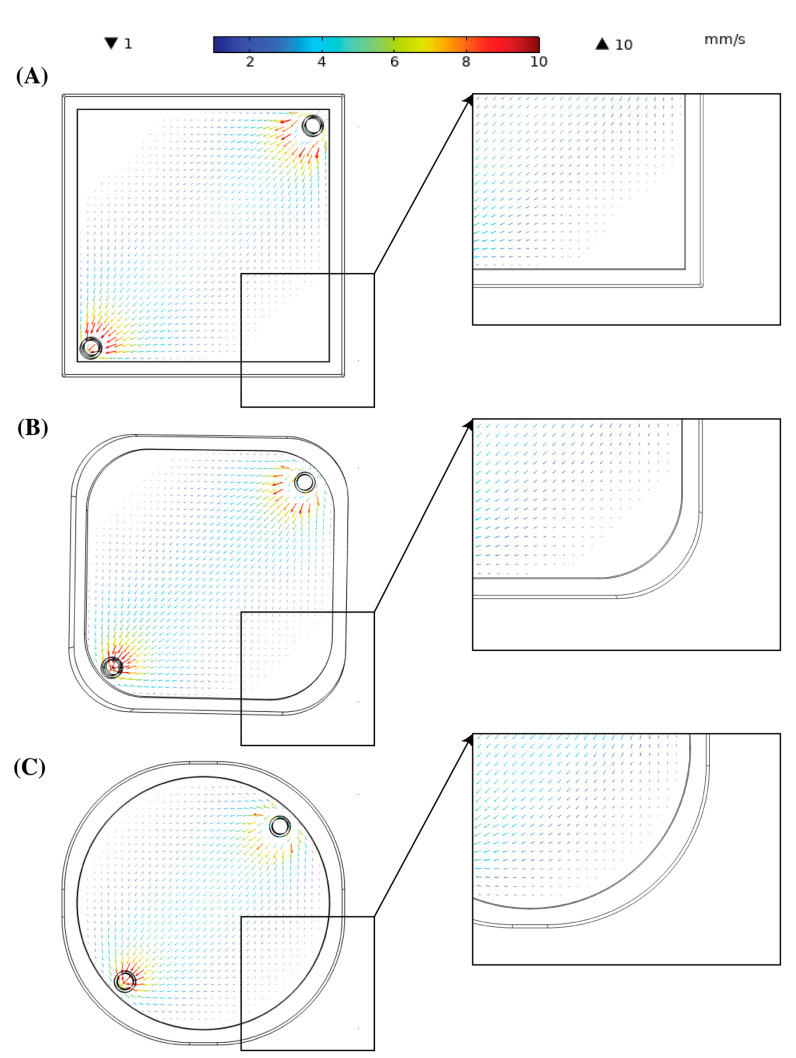
Velocity field of (**A**) Geometry 1, (**B**) Geometry 2, and (**C**) Geometry 3 at 0.25 L/min blood inflow. The arrow directions represent the direction of the blood flow, while the arrow length and color represent the magnitude of the vector. A scaling factor of 1000 was applied to the fields in (**A**–**C**) to improve the visibility of the arrows. The colors represent the true magnitude of the velocity. The blood inlet is located at the top right corner and blood outlet is located at the bottom left corner.

**Figure 5 bioengineering-10-00969-f005:**
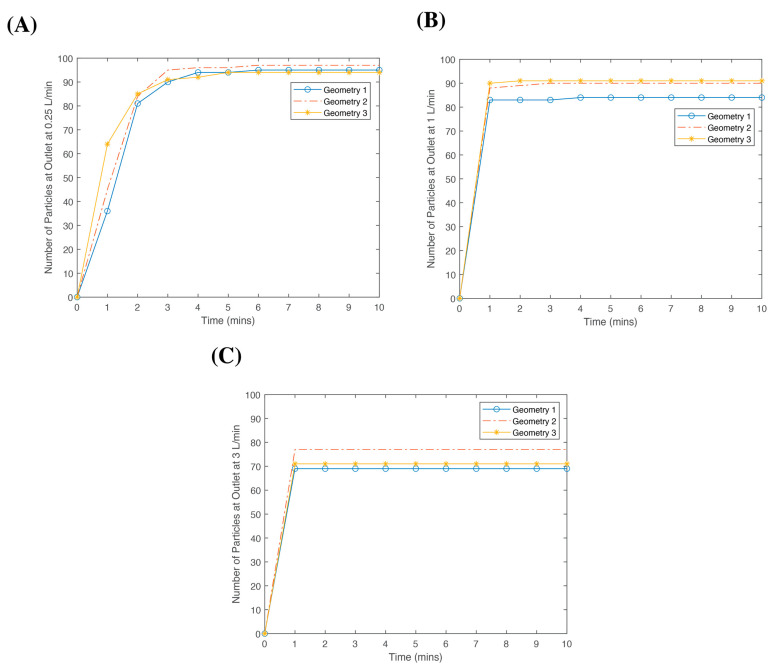
Total number of particles at the outlet for Geometries 1, 2, and 3 at the various flow rates: (**A**) 0.25 L/min 1; (**B**) 1 L/min; and (**C**) 3 L/min within 10 min of releasing 100 particles from the inlet.

**Figure 6 bioengineering-10-00969-f006:**
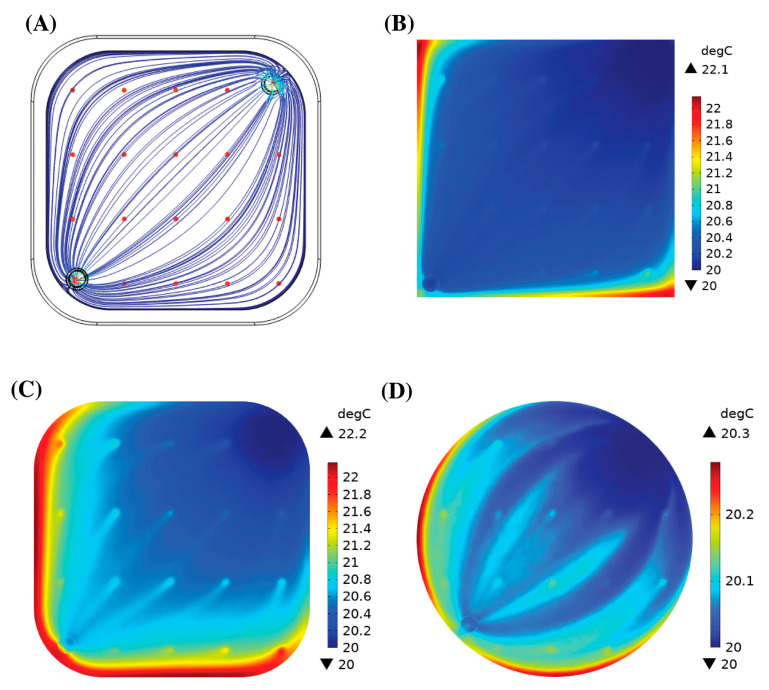
(**A**) Velocity streamlines with 20 LEDs at the top of the photo-ECMO device. (**B**) Temperature of the blood in Geometry 1 at a blood flow rate of 0.25 L/min. (**C**) Temperature of the blood in Geometry 2 at a blood flow rate of 0.25 L/min. (**D**) Temperature of the blood in Geometry 3 at a blood flow rate of 0.25 L/min. The blood inlet is located at the top right corner and blood outlet is located at the bottom left corner.

**Figure 7 bioengineering-10-00969-f007:**
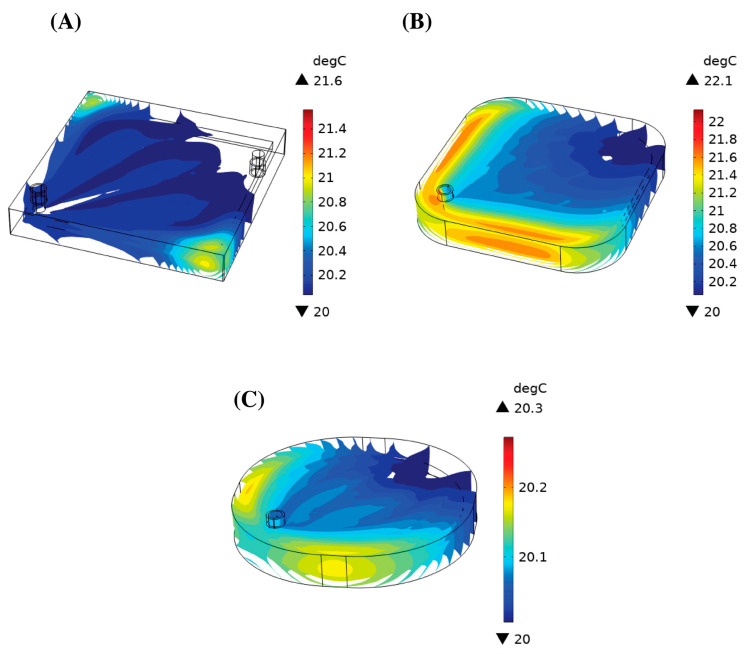
Iso-surfaces of the temperature in (**A**) Geometry 1, (**B**) Geometry 2, and (**C**) Geometry 3. The blood flowed in from the bottom right and out from the top left.

**Table 1 bioengineering-10-00969-t001:** Development of the photo-ECMO device, including the year of publication.

Year	Model Type	Oxygen Pressure Status of the Gas Compartment	Capable of Animal Rescue	Number of ECMO Devices	CO Half-Life in 1 L of Blood	Refernces
2019	Rat	Normobaric	Yes	1	36 ± 5.5 min	[25]
2021	Pig	Normobaric	Yes	6	6.3 ± 1.2 min	[24]
2021	In Vitro	Hyperbaric	N/A	4	5.2 ± 0.4 min	[26]
2023	Numerical Model	Normobaric	N/A	1	NA	[27]
2023	In Vitro and Mathematical Model	Normobaric	N/A	1	1.5 min	[28]

**Table 2 bioengineering-10-00969-t002:** Discretization of the modeling geometries.

Domain	Geometry 1	Geometry 2	Geometry 3
Blood compartment (without tubes)	Extra fine mesh with a maximum growth rate of 1	Extremely fine mesh with maximum element size of 2 mm	Extremely fine mesh
Blood in tubes	Extremely fine mesh	Normal mesh	Extremely fine mesh
Plexiglass casing	Extra fine mesh	Extremely fine mesh	Extremely fine mesh

**Table 3 bioengineering-10-00969-t003:** Grid sensitivity analysis for Geometry 1 at a flow rate of 0.25 L/min.

Plexiglass	Blood in Tube	Channel	Computation Time	Average Velocity in Photo-ECMO Device (mm/s)	Average Velocity in Tubes (mm/s)	Pressure Drop (Pa)	Max Temperature @ 5 min (C)	Number of Particles at Outlet
General physics, Extremely fine	Fluid Dynamics, Normal	Fluid Dynamics, Normal	Flow—00:04:26	2.98	32.86	6.4726	22.81	99
			Heat—00:38:05					
			Particle tracing—00:08:44					
General physics, Extremely fine	Fluid Dynamics, Finer	Fluid Dynamics, Finer	Flow—00:04:30	2.99	32.89	6.4538	22.81	97
			Heat—00:45:24					
			Particle tracing—00:08:04					
General physics, Extremely fine	Fluid Dynamics, Extremely Fine	Fluid Dynamics, Extremely fine	Flow—00:06:48	3	32.91	6.4340	22.83	95
			Heat—00:58:32					
			Particle tracing—00:09:52					

**Table 4 bioengineering-10-00969-t004:** Velocity and pressure drop in the photo-ECMO device.

	Blood Flow Rate (L/min)	Average Velocity in Photo-ECMO (mm/s)	Maximum Velocity in Photo-ECMO (mm/s)	Average Velocity in Inlet and Outlet Tubes (mm/s)	Pressure Drop (Pa)
Geometry 1	0.25	2.90	57.17	32.95	6.4025
	1	12.04	237.59	132.13	34.1191
	3	38.46	596.44	396.72	240.9999
Geometry 2	0.25	2.74	57.02	32.87	5.3801
	1	11.57	226.34	131.77	30.1926
	3	36.42	558.37	395.44	210.3300
Geometry 3	0.25	2.80	56.82	32.92	4.9975
	1	11.97	235.67	131.87	29.1249
	3	39.29	585.71	395.73	208.1257

**Table 5 bioengineering-10-00969-t005:** Temperatures in the photo-ECMO device.

Blood Flow Rate (L/min)	Maximum Temperature in Photo-ECMO Device after 5 Minutes (°C)	Maximum Temperature in Photo-ECMO at a Steady State (°C)	Average Temperature in Photo-ECMO at a Steady State (°C)	Average Temperature at the Blood Outlet at a Steady State (°C)
0.25	21.83	21.59	20.07	20.08
1	21.95	23.99	20.77	21.05
3	22.3	22.17	20.22	20.27
0.25	20.83	22.19	20.58	20.91
1	21.94	20.86	20.16	20.23
3	21.68	20.68	20.05	20.07
0.25	20.57	20.28	20.04	20.06
1	21.57	22.53	20.60	20.80
3	21.32	20.93	20.16	20.20

## Data Availability

The data presented in this study are available on request from the corresponding authors.

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
