# Peer review of "Design Optimization of a Phototherapy Extracorporeal Membrane Oxygenator for Treating Carbon Monoxide Poisoning"

_bioengineering, 2023, doi:10.3390/bioengineering10080969_

Round 1
Reviewer 1 Report
The authors performed a numerical study on Phototherapy Extracorporeal Membrane Oxygenator for Treating Carbon Monoxide Poisoning.
The paper is generally well prepared and has a good scientific soundness. It can be accepted for publication after addressing the following points:
Why are the equations related to light propagation not presented?
The energy equations in the solid domains are to be presented.
A grid sensitivity test is to be performed.
A validation/verification of the numerical model is to be performed.
What is the used convergence criterion?
What is the used time step?
The thermophysical properties of the Blood, Tubes and Plexiglass are to be presented.
For the blood, have you considered temperature dependent or independent thermophysical properties?
Information about the characteristics of the used computer and the computational time are to be presented.
For a better understanding of the flow structure, it will be interesting to present the 3D streamlines. Similarly, it will be interesting to present the 3D iso-surfaces of temperature.
Reviewer 2 Report
It is an interesting paper with great potential for clinical application. Nevertheless, your approach remains partially inconclusive considering that you have not modelled CO removal as you rightly stated in the discussion. Are you proposing Geometry 2 as an alternative in terms of optimisation purposes? I think the discussion should be more focused on your results and your views in terms of further developments for clinical application. It would also be appropriate to discuss your previous experimental work, which would add value to your proposal and strengthen your argument. How are you planning your next step? How confident are you with your current approach? Are you ready for an initial clinical trial? If not, what are the additional issues and limitations to address? These are all points to develop and argue in the discussion in order to deliver your message in a more impactful manner.
Here are some other comments and suggestions.
Line 42: it is more appropriate to say “altered mental status”.
Line 126: it is more appropriate to say “clinically not feasible”.
Line 128: surely you meant cm3; cm-3 does not make sense.
Line 142-143: it would be more appropriate to say “Blood flows within the chamber around the hollow fibres; however, the fibres are not modelled in this study for the sake of computational simplicity.”
Line 246: it would be more appropriate to say “...of deposited energy in the blood...”
Line 274: it is more appropriate to say “Additionally, some areas have no arrows showing...”
Line 278-280: it would be more appropriate to say “The pressure drop is similar at the different flow rates and geometries although Geometry 1shows a higher pressure drop (30 Pa) at 3 L/min than Geometry 2 and 3.”
Line 296-299: it would be more appropriate to keep the past tense all of the time as follows “The number of particles at the outlet did not change for all geometries and flow rates after 5 minutes. Nevertheless, the number of particles at the outlet for Geometry 1 and 2 increased by one at 0.25 L/min after 6 minutes after which it remained the same.”
Line 351: it is more appropriate to say “Due to the depth of light penetration in the blood, ...”
Line 354-361: it should be “cm3” for all the quantities, shouldn’t it?
Line 423-424: it would be more appropriate to say “The effectiveness of this approach for CO removal remains unknown because CO removal was not modelled.”
The use of the English language is appropriate although some re-polishing is required.
Round 2
Reviewer 2 Report
You have addressed all my comments and suggestions.
The paper is certainly much stronger now
I have only one observation to make.
Page 7, line 211: you consider a time step of 0.01 min. Is this what you did or did you mean a time step of 0.01 s?
A time step of 0.01 min would be 0.6 s, which is completely different.
Can you clarify, please?
Author Response
Dear Reviewer,
Thank you for the observation. The time step was in minutes (0.001 mins) and we have recalculated it in seconds (0.06 s). This has been updated in the manuscript.